# Behavioral Coding of Captive African Elephants (*Loxodonta africana*): Utilizing DeepLabCut and Create ML for Nocturnal Activity Tracking

**DOI:** 10.3390/ani14192820

**Published:** 2024-09-30

**Authors:** Silje Marquardsen Lund, Jonas Nielsen, Frej Gammelgård, Maria Gytkjær Nielsen, Trine Hammer Jensen, Cino Pertoldi

**Affiliations:** 1Department of Chemistry and Bioscience, Aalborg University, Frederik Bajers Vej 7H, 9220 Aalborg, Denmark; jnielc21@student.aau.dk (J.N.); maniek21@student.aau.dk (M.G.N.); cp@bio.aau.dk (C.P.); 2Aalborg Zoo, Mølleparkvej 63, 9000 Aalborg, Denmark

**Keywords:** machine learning, nocturnal behavior, computer vision, captive elephants

## Abstract

**Simple Summary:**

This paper presents a way to automate computer vision processes applied to behavior recognition on closed-circuit television (CCTV) footage of two captive African elephants. Object detection software using both Create ML and DeepLabCut was used to control the accuracy of using such models, and those models were subsequently used to analyze seven days’ worth of nighttime footage to assess the general behavioral patterns of the elephants, showcasing the possibility of using automated tools for behavioral analysis.

**Abstract:**

This study investigates the possibility of using machine learning models created in DeepLabCut and Create ML to automate aspects of behavioral coding and aid in behavioral analysis. Two models with different capabilities and complexities were constructed and compared to a manually observed control period. The accuracy of the models was assessed by comparison with manually scoring, before being applied to seven nights of footage of the nocturnal behavior of two African elephants (*Loxodonta africana*). The resulting data were used to draw conclusions regarding behavioral differences between the two elephants and between individually observed nights, thus proving that such models can aid researchers in behavioral analysis. The models were capable of tracking simple behaviors with high accuracy, but had certain limitations regarding detection of complex behaviors, such as the stereotyped behavior sway, and displayed confusion when deciding between visually similar behaviors. Further expansion of such models may be desired to create a more capable aid with the possibility of automating behavioral coding.

## 1. Introduction

### 1.1. Objectives of Wildlife Conservation

The World Association of Zoos and Aquaria (WAZA) aims to conserve endangered species through breeding programs and exchange of captive animals. This association requires certain standards of its members and emphasizes the importance of welfare among captive animals [1]. Based on consistent and current research, good physical and psychological welfare among captive animals must be maintained and therefore associations such as this play an important role in conservation [1,2,3,4].

Because of the importance of animal welfare, it is essential to ensure that captive animals are consistently studied in relation to their behavioral reactions to different aspects of their captive lives, such as enclosure design, enrichment, and much more [5,6,7,8]. An example of an animal that may require such studies to properly conserve the species is the African elephant (*Loxodonta* sp.), which has faced great declines in population size all over Africa [5,9].

### 1.2. Captive Elephant Behavior and Welfare

To accommodate the welfare needs of captive elephants, normal behaviors must first be monitored and understood [6,10,11]. Behaviors such as foraging, locomotion, social behavior, etc., likely influence the welfare of elephants and help understand undesired behaviors that may indicate stress [12,13,14]. It is important to also address the nocturnal behavior of elephants since night behavior can differ from behaviors observed during the day. An example of one such behavior is recumbent sleep, where the elephants lay down and sleep, which exclusively occurs during the night [14,15]. On average, captive African elephants lie down to sleep around two hours per night and tend to lie down more if their bedding is comfortable [14,15,16,17]. Besides rest, feeding and atypical behaviors are the behaviors most commonly observed in captive elephants, yet the activity level is lower at night compared to during the day [14,18].

Atypical behaviors observed in animals kept in captivity are usually those that deviate from the norm for their species and are not commonly observed in their natural habitat. These behaviors are frequently regarded as signs of compromised welfare [19,20,21,22]. A type of stereotypic behavior is characterized by the consistent and inappropriate repetition of specific movements or body postures. These actions seemingly lack any purpose or function and appear to be coping mechanisms to reduce stress, but the exact causes remain unclear [21,22,23,24,25]. Different forms of stereotypic behavior have been observed in elephants, including whole-body movements [10,13,22,26,27]. Among whole-body movements, ‘swaying’ is most common and is defined as a rhythmic side-to-side movement of the body, typically observed while standing [12,18,22]. Consistent observations of elephants may be helpful in handling these behaviors when they arise.

### 1.3. Machine Learning as a Tool for Behavioral Analysis

One widely used method for observation of animal behavior is videography [14,28], although this method can be highly time-consuming [29,30,31,32]. Manual scoring is limited by human capabilities, such as the observer not recognizing behavioral patterns or failing to spot new patterns. At the same time, it is difficult to standardize the scoring of behaviors by human observers due to subjectivity [33]. Inconsistency between different observers can therefore not be avoided completely [31]. Furthermore, it is challenging to track multiple animals and behaviors at the same time, despite the use of video material [29,31]. Some of these logistical problems with behavioral analysis may be aided with the use of machine learning [32].

Machine learning used in video and image analysis (computer vision) has been explored in recent years in application to a variety of purposes [34,35]. The use of object detection as a machine learning tool to find and recognize a given object has been investigated previously and used to recognize animals and their behaviors [32,36,37,38]. Tools such as this may prove useful as a way of automating behavioral analysis in the near future, which may reduce the workhours required of the researcher [30,32]. However, these uses and methods are still in their infancy which necessitates further investigation of different machine learning models, methods, and implementations [36,37,39].

DeepLabCut is a Machine Learning software that specializes in pose estimation in video material, by using points for tracking specific body parts [40,41]. This is utilized in behavior tracking by marking body parts of interest on a relatively small dataset of images showing a diverse range of behaviors by the subject of interest [32]. Constructing a DeepLabCut model capable of accurately tracking body parts of interest may allow for automatization of behavior coding in behavioral studies [42].

Create ML is a built-in application for iOS products with the ability to train custom machine learning models such as object detection models with no code. Create ML models can be trained to detect and recognize objects of interest, such as an elephant executing a specific behavior, by annotating a relatively small and diverse dataset. This annotation can be used in various ways, such as using the image annotation tool RectLabel for marking boxes, polygons, or skeletons on the subject, and categorizing the behavior. This may allow for construction of a simple model that can recognize simple behaviors with relatively user-friendly software [43].

### 1.4. Aim of This Paper

This paper aims to use DeepLabCut and Create ML to construct models capable of tracking selected body parts and classifying elephant behaviors, aiming to streamline and automate behavioral analysis processes, thus ultimately alleviating the workload of researchers and zookeepers, and standardizing behavioral coding. This study simultaneously examines nocturnal activity of two captive African elephants with the following hypotheses:This study expects that the machine learning models can predict selected behaviors on the same level as manual scoring;This study expects that behavioral differences between the elephants and behavioral differences between days can be demonstrated using selected computer vision models.

## 2. Materials and Methods

### 2.1. Subjects and Enclosure

The behavior of two captive female African elephants, exhibited in Aalborg Zoo, Denmark, was examined. Both elephants were born wild in South Africa around 1982 and relocated to Aalborg Zoo in 1985. In this study, the elephants are referred to as Subject A and B.

The elephant enclosure consisted of an indoor area and an outdoor area. The elephants did not have access to the outdoor enclosure at night during the examined period. The indoor enclosure consisted of concrete floors and walls with metal wires towards the visitor area (see Appendix A). The two elephants were able to have physical contact during the night through metal bars between enclosures E1 and E2. Subject A had access to enclosure E1, and a corridor attached to the enclosure (56 square meters). Subject B had access to enclosures E2 and E3 as well as a corridor measuring a total of 116 square meters.

The elephants’ diet consisted of branches, seed grass hanging from nets in the inside and outside enclosures, and concentrate pellets that would periodically be released into the enclosure from a timer-automated mechanism, as a type of enrichment. Fresh fruits and vegetables were spread around their outdoor enclosure daily, which allowed for foraging behaviors. Foraging boxes, accessible with use of their trunks at the back wall of their enclosure, were also opened periodically throughout each night, controlled by a timer.

### 2.2. Data Collection

Prior to data collection, an ethogram with selected behaviors was made. The behaviors were determined based on similar behavioral studies of the same elephants in previous publications, namely Bertelsen et al. (2020) and Andersen et al. (2020), and were modified for the purpose of this study [14,44]. This ethogram was used to conduct a manual control, using researchers experienced in behavioral coding, where the researchers were assigned a subject each. The behaviors were coded continually on a second-by-second basis. These scorings were used to compare manual coding with the models; see Table 1.

The data were collected from the 12th to 18th of March 2024 to use for model creation, and from the 16th to the 22nd of April 2024 for analysis purposes, using three cameras (ABUS, 25 FPS). The three cameras were placed at the visitor viewing side facing towards each enclosure (see Appendix A). The two elephants were observed during the night for seven hours from 22:00 to 05:00 (DST).

### 2.3. Data Analysis

The analyzed days in this experiment provided data which were compared with statistical tests using Excel (Version 2404 Build 16.0.17531.20120) and RStudio (R version 4.1.2 (1 November 2021)).

For body part tracking, DeepLabCut (version 2.3.9.) was used [32,42]. Specifically, 244 frames taken from 70 30-min videos were labelled (95% were used for training), and no preprocessing was performed. A ResNet-50-based neural network was used with the parameters set to 400,000 training iterations. Validation was carried out with a single shuffle, and the test error found was 17.44 pixels, train 2.4 pixels (image size for creating the model was 1920 by 1080). A p-cutoff of 0.5 was used to condition the *x*- and *y*-coordinates for future analysis. DeepLabCut does not provide annotations of behavior, and it is required of the user to manually define and interpret the coordinates of the results. In this study, the behaviors were classified in Excel using parameters set by comparing the coordinates of relevant body parts with the manually recorded data and videos from the control period (April 16). Each parameter consisted of distinct coordinate limits and requirements to fit the ethogram, resulting in frame-by-frame behavioral coding. The locations of the selected behaviors can be seen in Appendix B. The algorithm for classifying behaviors using DeepLabCut can be seen below in Algorithm 1.
**Algorithm 1.** **DeepLabCut operation****Input**: Video containing the subject to be tracked**1. Load dataset.** Load the video into DeepLabCut;**2. Define keypoints.** Specify keypoints of interest (body parts like head, tail, limbs);**3. Annotate frames.** Annotate a subset of frames manually by marking the keypoints;**4. Model training.** Use annotated frames to train the pose estimation model;**5. Pose estimation.** Apply the trained model to new video material;**6. Refine model (optional).** Correct predictions and retrain the model;**7. Process coordinates.** Extract CSV file and filter the coordinates for desired body parts;**8. Classify behaviors.** Set limits for each coordinate corresponding to a desired behavior and filter frames that fulfill the criteria.**Output:** Subset of data points that can be classified as a specific behavior.

Difficulties arose with defining some of the parameters, such as ‘Drinking’, which proved undefinable for both subjects as the label on the trunk tip was unstable. Furthermore, no distinct parameter was definable for the behavior ‘Hay-net’ for Subject B as the foraging box and the hay net were located at approximately the same place in the video frame.

For object detection, Create ML (version 5.0 (121.1)) was used [43]. Specifically, 370 image frames were extracted from the model creation period and annotated using bounding boxes in RectLabel Pro (version 2023.11.19). Each frame was annotated with a selected behavior as seen in Table 1. ‘Swaying’ was not labelled for the object detection part. The behavior ‘Standing’ was labelled 232 times, ‘Foraging’ was labelled 42 times, ‘Lying down’ was labelled 39 times, ‘Drinking’ was labelled 39 times, and ‘Hay-net’ was labelled 18 times. The dataset was split into two sets, one containing images for training and one containing images for validation. The model was trained with 6000 iterations, and the training set had an accuracy of 95% whilst the validation set had an accuracy of 75%. Create ML automatically classifies the behavior using bounding boxes resulting in a frame-by-frame behavioral coding output. The algorithm for classifying behaviors using Create ML is seen below in Algorithm 2.
**Algorithm 2.** **Create ML operation****Input**: Video containing the subject to be tracked.**1. Load dataset.** Load the video into RectLabel and extract images;**2. Define bounding boxes.** Specify categories of each bounding box of interest (behaviors, such as lying down or standing);**3. Annotate frames.** Annotate a subset of frames manually by drawing bounding boxes;**4. Model training.** Use annotated frames to train the model;**5. Pose estimation.** Apply the trained model to new video material;**6. Refine model (optional).** Correct predictions and retrain the model;**7. Process coordinates.** Extract CSV file containing frames annotated with behaviors.**Output:** Dataset containing the predicted behaviors at all analyzed frames.

To appraise the accuracy of the models designed with Create ML and DeepLabCut, a control was analyzed manually from the video footage from the 16th of April and used to compare the models’ results. To test the accuracy of the models compared to the control, a confusion matrix was conducted. In this case, a multiple class confusion matrix was produced and analyzed [45]. This will give insight into where the model performs well and has a high accuracy, as well as where mistakes occur, such as when the model mislabels a behavior. The columns of such a matrix represent the manually observed behavior of the individual while the rows represent the predicted behavior from the models. Time budgets and cumulative graphs were also used to further appraise the models [46]. Furthermore, Kendall’s Coefficient of Concordance was used to measure the agreement between the models and the control [47].

The behavioral analysis consisted of time budgets, cumulative graphs, and Kendall’s Coefficient of Concordance between days. Time budgets were made for each elephant each day and for the whole study period. This was carried out using the sums, transformed into percentages, of observed time spent on each behavior, where the out of view percentage made up the time where no behavior was observed or classified. Time budgets for each day were used to investigate daily differences in behavior. The time budgets for the whole study period were used to see how much time was spent on each behavior in total and to compare the different models with the control. Kendall’s Coefficient of Concordances were used on the data from the time budgets to analyze the similarity between the observed behavior between different days.

Cumulative graphs were made for each behavior each day. The graphs were made with both the manually recorded data and the model data for the control day, while the rest of the study period only had cumulative graphs made for the Create ML model.

A Spearman rank correlation test was used to investigate correlations between the subject’s ‘Lying Down’ behavior from night to night. Correlations were also used to test the similarities between the subjects and if they exhibited similar behavioral patterns throughout the night.

The possibility to observe the stereotypic behavior ‘Swaying’ was also examined. This was accomplished by calculating the Euclidian distance between a given point of the trunk root, labelled by the DeepLabCut model, and the succeeding point [48]. The distances between the points were calculated and plotted as a cumulative graph together with the actual cumulative time spent on the sway behavior, so that sway could be observed as steep increases in the cumulative sum.

## 3. Results

### 3.1. Comparability of Manual and Automatic Behavioural Observations

#### 3.1.1. A General Overview

First, the capabilities of the two machine learning models, compared to a manually conducted analysis using an ethogram, have been displayed using time budgets, illustrated below in Figure 1.

The time budgets for both subjects, using both models, showed similar percentages to the manual observations regarding standing and lying down with relatively small differences. For Subject A, both models had a 10% out of view percentage. Since the manual observations showed an out of view percentage close to zero, this indicates that these were caused by uncertainty by the models, causing them to not label the subject. Both models for Subject A also displayed a lower percentage for ‘Foraging’ than the manual observations. For Subject B there were notable differences in ‘Foraging’ and ‘Hay-net’, which is likely to be caused by the hay-net being close to the foraging boxes, thus showing overlapping coordinates when observed by the DeepLabCut model. The Create ML model also had a lower ‘Hay-net’ percentage, but contrarily a higher ‘Foraging’ percentage than the manual observations. The out of view percentage for the DeepLabCut model was noticeably higher than the Create ML model, considering that the manual observations showed an out of view percentage close to zero. This was likely also caused by lack of labelling by the model. ‘Drinking’ was left out of the DeepLabCut model due to limitations in defining the parameters of this behavior, caused by a lack of consistent appropriate labels needed to properly categorize the behavior.

To further investigate the similarities between the machine learning models and the manually conducted observations, cumulative graphs for each subject, showing each behavior tracked with each method, have been constructed and displayed below in Figure 2.

The cumulative graph for Subject A showed a lower sum but similar shape for ‘Lying down’, which indicates that a period of observations of this behavior went unlabeled for both models. ‘Standing’ had a similar shape for all methods and a very similar sum for the DeepLabCut model, whereas the Create ML model had a higher sum. ‘Foraging’ had similar shapes for all methods; however, the sums were generally lower for the models. ‘Hay-net’ and ‘Drinking’ were difficult to distinguish clearly, due to low values.

The cumulative graph for Subject B had very similar values for ‘Lying down’ for all methods. All methods showed similar shapes for ‘Standing’, although the sums were lower for both models, most noticeably for the DeepLabCut model. ‘Foraging’ also had similar shapes but higher and lower sums for the Create ML model and DeepLabCut model, respectively. ‘Hay-net’ also showed a somewhat similar shape to the manual observations for the Create ML model, but this model had a lower sum. The DeepLabCut model showed a ‘Hay-net’ sum close to zero, due to difficulty in defining these parameters in the enclosure.

Similarly to the time budgets, ‘Drinking’ was left out of the cumulative graphs for the DeepLabCut model due to limitations in defining the parameters. Furthermore, the shapes in ‘Lying down’ for both subjects seem largely different; however, this is caused by the chosen type of cumulative graph.

#### 3.1.2. Investigating the Reliability of Two Machine Learning Models

To investigate the reliability of the two models, Kendall’s Coefficient of Concordance was utilized. The concordance (W-value) between the models and the control was found to be 0.85 with a *p*-value of 0.026 for Subject A, and 0.90 with a *p*-value of 0.019 for Subject B. This indicates a high concordance between the models that is not stochastic for both subjects. This high concordance means that the models and the manual scoring mostly agree on the observed behavior. However, this concordance is not perfect, so a slight disagreement is present.

To test the accuracy of the models, a confusion matrix for the control period was made and the models were compared to the manually observed values and normalized (Appendix C). As seen in Table A1, Table A2, Table A3 and Table A4, both models predicted highly similar values for the behavior ‘Lying Down’ for both subjects compared to the manually observed values. For the DeepLabCut model, the predicted values for ‘Standing’ and ‘Foraging’ were highly similar to the observed values for Subject A; but for Subject B, the model was more inaccurate. The opposite applies for Create ML, where the model was generally most accurate for Subject B. For the behavior ‘Hay-net’, the DeepLabCut model struggled to predict the correct behavior for Subject A, and for Subject B the classification parameters were not defined, and therefore no value was predicted for this behavior. The Create ML model for the behavior ‘Hay-net’ predicted highly similar values for Subject A, but for Subject B the behavior was often misclassified as ‘Foraging’. For the behavior ‘Drinking’, the Create ML model often predicted the behavior as ‘Standing’ for both subjects and for the DeepLabCut model the classification parameters were not defined and therefore no values were predicted correctly. Finally, the predicted values by both models for out of view for Subject B were highly similar to the observed value but it is notable that the total manually observed value for this behavior is 17 s out of 7 h of observation time and is therefore arguably negligible.

### 3.2. Using Machine Learning Models for Behavioural Analysis

#### 3.2.1. Assessing Behavioral Differences

To analyze behavioral differences between the two subjects, two time budgets for the total sums of each behavior for seven nights were constructed for both machine learning models, as is seen below in Figure 3.

The time budgets display some differences between the models, especially noticeable in the out of view percentages for Subject B. ‘Standing’ and ‘Lying down’ were similar for both models for both subjects. ‘Foraging’ was similar for Subject A in both models but was slightly higher for the Create ML model for Subject B, possibly due to the lower out of view percentage. The DeepLabCut model did not measure the ‘Drinking’ behavior for either subject. Comparing the total time budgets for the period between the two subjects only showed slight differences.

To further investigate the behaviors of both subjects during the observed period, the sums of behaviors for all individual days have been shown as time budgets in Appendix D. The time budgets for Subject A showed some variation in the behaviors, especially in ‘Foraging’. ‘Standing’ and ‘Lying down’ also varied somewhat from night to night. Subject B also showed variation in ’Foraging’, but generally less so than Subject A. ‘Standing’ and ‘Lying down’ varied somewhat for Subject B. There was a noticeable difference in out of view percentages for Subject B, depending on the model, with a consistently much lower percentage for Create ML. The behavioral differences were examined further using cumulative graphs (Appendix E).

Kendall’s Coefficient of Concordance was used to examine if the amount of time spent on each behavior was the same each day. The analysis was conducted on the results of both the DeepLabCut and the Create ML model. Subject A showed a concordance of 0.935 with a *p*-value of $4.29\times 10^{-6}$ for the DeepLabCut model and 0.865 with a *p*-value of $1.31\times 10^{-5}$ for the Create ML model. Subject B showed a concordance of 0.951 with a *p*-value of $3.3\times 10^{-6}$ for the DeepLabCut model and 0.869 with a *p*-value of $1.21\times 10^{-5}$ for the Create ML model. All the concordance values were high with a significant *p*-value indicating that the high concordance is not stochastic. This high concordance indicates an agreement in the observed time a subject spends on different behaviors from day to day. This concordance is not perfect meaning some variations are still present in the subjects’ nocturnal behavior.

Spearman’s rank correlation for the behavior ‘Lying down’ was investigated for both models, and the analysis was split between days and individuals (Appendix F).

The analysis between days resulted in mainly positive correlations. Subject A had correlations between 0.965 and −0.053 for DeepLabCut and 0.970 and −0.135 for Create ML. Negative correlations were observed between the 20th and 22nd of April for DeepLabCut and the 19th and 20th of April for Create ML. Subject B had correlations between 0.999 and 0.019 for DeepLabCut and 1.000 and 0.311 for Create ML. No negative correlations were found for Subject B.

The analysis between the two subjects also resulted in mainly positive correlations (Appendix G). The results of the DeepLabCut model had correlations between 0.975 and −0.361. A single negative correlation was found between the 20th of April for Subject A and the 19th of April for Subject B. The results of the Create ML model had correlations between 0.977 and 0.052. No negative correlations were found for the Create ML model.

#### 3.2.2. Investigating Further Applications of Automatic Behavioral Coding

Certain behaviors of a more complex character may potentially be assessed using machine learning methods for behavioral coding. One such behavior is the stereotyped behavior ‘Swaying’, which is largely relevant for elephants [14]. This behavior is difficult to categorize simply, as has been carried out for the previously mentioned behaviors, since swaying can happen anywhere in the frame and is primarily observable through a side-to-side motion of the elephant’s trunk and head. To visualize this behavior using data from the DeepLabCut model, the cumulative distance moved by the point labelled at the trunk root of Subject B was plotted, along with the actual sway noted manually in the control period as a cumulative graph in Figure 4.

As seen in the cumulative graph, a steep increase in the trunk distance appears around 01:15, which approximately matches with the manually observed sway behavior occurring at this time. This is because a steep increase in distance moved by the trunk root will occur as a result of the sway behavior.

## 4. Discussion

### 4.1. Performance and Limitations of the Two Machine Learning Models

Before using the two constructed machine learning models, it must first be investigated how accurate they are compared to manually recorded observations. The concordance test between the models showed a high agreement between the models and manual observations (W = 0.848), although there is seemingly room for improvement. The confusion matrixes for each model compared to the manual observations also displayed high accuracy in detecting some behaviors, such as ‘Standing’ and ‘Lying down’, although with certain challenges, such as confusing the ‘Hay-net’ behavior with ‘Foraging’ for Subject B. This suggests that there may be a need for improving the parameters of classifying each behavior or training the models with better or more material to account for different environments, footage qualities, and a broader range of behaviors. Mathis et al. (2018) discussed the capabilities and limitations of DeepLabCut for various behaviors, noting similar challenges in behavior recognition and out of view instances [32].

It must, however, be noted that the accuracy of these models is based on comparison with manual observations, which itself has inherent problems. Manual observations are not entirely accurate, since they may lack precision in noting the exact time a behavior takes place, and there may be differences in how a sequence of behaviors is coded by different researchers, which usually necessitates inter-rater reliability tests [41,49,50]. These issues should not be present in machine learning models since a behavior is coded at the exact frame and can be standardized across studies. The capabilities of models constructed in both Create ML and DeepLabCut thus emphasize the potential of machine learning models to complement and enhance traditional manual observations in behavioral studies. Further optimization of such computer vision models may also include image processing such as exploring different color spaces and image augmentations [51,52].

### 4.2. Nocturnal Behavioral Differences of the Two Subjects

The two machine learning models were used to automatically observe the two subjects of the study, with the aim of assessing whether the nocturnal behaviors varied between each subject and individually across each observed night. This analysis was carried out using time budgets, cumulative graphs, and correlations.

Firstly, the total time budgets for each subject across all nights displayed only slight differences between the subjects, most notably in foraging behavior, which might be higher for Subject B. This slight difference is supported by the high concordance values between the days, meaning the observed behavior only differs slightly from day to day. It is, however, inconclusive whether Subject B generally carries out more foraging behavior, due to the differing out of view percentages caused by the lack of confidence by the models. With a closer look at the behavioral patterns using the cumulative graphs (Appendix E), it does, however, appear that the two subjects have some differences. Once again, it appears that ‘Foraging’ is generally higher for Subject B, along with ‘Drinking’. From the ‘Lying down’ cumulative graph it also appears that sleeping patterns may be somewhat different, since Subject A appears to wake up and walk around more commonly throughout the night, whereas Subject B appears to be lying down for longer periods at a time. The correlations calculated regarding the sleeping patterns compared between the two subjects also showed some correlation, indicating that the subjects go to sleep at similar times, although this varies each night. This shows that the two elephants differ from each other in the nocturnal behavioral patterns; however, there do not appear to be large differences, and overall, the subjects typically carry out somewhat similar behavioral patterns throughout the night. This is in accordance with Bertelsen et al. (2020) which studied the same subjects and found some personality differences displayed through behavior, but similarly the differences were relatively small [14]. A study by Rees (2009) also found behavioral differences on an individual basis in captive Asian elephants (*Elephas maximus*) [10]. This is similar to Tobler (1992), Holdgate et al. (2016), and Schiffmann et al. (2023) who examined recumbent sleep behavior in zoo-housed Asian and African elephants, and also found differences on an individual basis [15,17,53].

It was investigated whether the individual subjects differed in their behavioral patterns from night to night, using time budgets for each night, along with cumulative graphs and correlations of their sleeping behavior. From the time budgets, both subjects appear to vary from night to night in all behaviors. ‘Foraging’ ranges from very low percentages (1–2%) to high percentages (18–22%), which is also apparent from the cumulative graph. This is in accordance with the study by Finch et al. (2021) who found varying feeding behavior in their nocturnal activity budgets for zoo-housed Asian elephants [54]. ‘Standing’ and ‘Lying down’ for both subjects also differed across nights with a range of approximately 20% difference for both behaviors. Further investigation of sleeping patterns using Spearman rank correlations showed that most days had very highly correlated values, indicating a general circadian rhythm; this is in accordance with a study by Casares et al. (2016) that investigated cortisol levels to establish the circadian rhythm of African elephants [55]. However, some days showed much weaker correlation, suggesting differences in night-to-night sleeping patterns caused by the elephants going to sleep at different times. This confirms the hypothesis that computer vision models are capable of demonstrating that the nocturnal behavioral patterns differ from night to night for both individuals, although there is some uncertainty of exactly how much the behaviors differ, due to the out of view percentages. This result is, however, in accordance with the studies by Rees (2009) and Holdgate et al. (2016) who found considerable day-to-day variation in activity budgets for a group of captive Asian elephants and for African and Asian elephants, respectively [10,15].

This study showed that the subjects were lying down approximately 35–39% of the observed time, or just over 2.5 h until 5:00, at which point they would still be lying down, as is seen in the cumulative graphs. This is in accordance with studies such as Holdgate et al. (2016) and Schiffmann et al. (2023) who found that elephants in captivity generally tend to lie down for a similar amount of time during the night; although they also note that the elephants may not be sleeping throughout all of this time [15,17].

### 4.3. Other Applications of Machine Learning Models for Behavioral Coding

As was displayed in the results regarding the sway behavior, it may be difficult to address complex behaviors, even though it may be possible through different techniques. One such technique was displayed by plotting the distance moved by a point on the trunk root of Subject B. The steep incline on the graph largely matches with the manually observed sway during the night, which indicates that using such a measurement might be useful for observing ‘Swaying’. Currently, the presentation of this behavior is, however, primarily visual since it may provide further challenges to precisely define the parameters capable of properly discerning when the gathered data should be categorized as sway behavior. Other challenges related to addressing complex behaviors, such as stereotypic and obsessive self-grooming in primates, face similar challenges since this behavior is also classified by a consistent repetition, which a computer vision model would have difficulties identifying on a frame-by-frame level. However, a study by Yin et al. (2024) is somewhat successful in showing distinct motion trajectories exhibited by a variety of different animals including tigers (*Panthera tigris*), bears (*Ursidae*), and wolves (*Canis lupus*), and classifying this as stereotypic behavior by assessing repetitive patterns [56]. Tackling issues that may arise using computer vision for behavior recognition is somewhat a case-by-case problem, where camera settings, image processing, and other issues should be considered, in order to fit the models appropriately to the research. However, developing such parameters and applying them to similar machine learning model data in the future would prove useful to quickly and accurately find stereotyped behavior to gain insight into the welfare of individual animals.

## 5. Conclusions

It is apparent from this study that using machine learning models from DeepLabCut and Create ML provides a capable tool for aiding or even replacing certain aspects of behavioral studies. The models could detect simple behaviors with high accuracy, although limitations were met when assessing repetitive behaviors such as ‘Swaying’. Similar complex behaviors, such as certain stereotypic behaviors, in other animals may prove equally challenging and the detection of it may require further work to be adequate.

Applying the models to seven nights of footage of nocturnal behavior provided general insight into behavioral patterns and differences between the two studied subjects, as well as differences between individually observed days. This showed that the constructed computer vision models can effectively aid in behavioral analyses, and further expansion and adjustments may be desired. This could potentially be achieved through exploring image augmentation, classification of complex behavioral patterns, or implementing such models to be readily available as a tool for zoological gardens.

## Figures and Tables

**Figure 1 animals-14-02820-f001:**
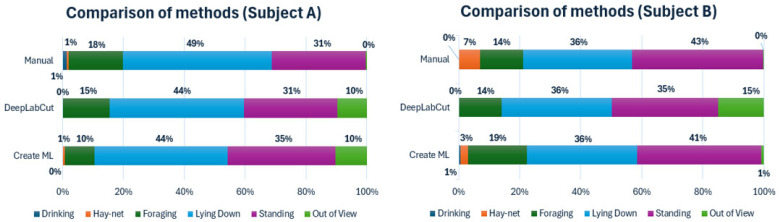
Time budgets for Subjects A and B of the 7 h control period, comparing manual observations with a DeepLabCut and a Create ML model. ‘Drinking’ is excluded for the DeepLabCut model.

**Figure 2 animals-14-02820-f002:**
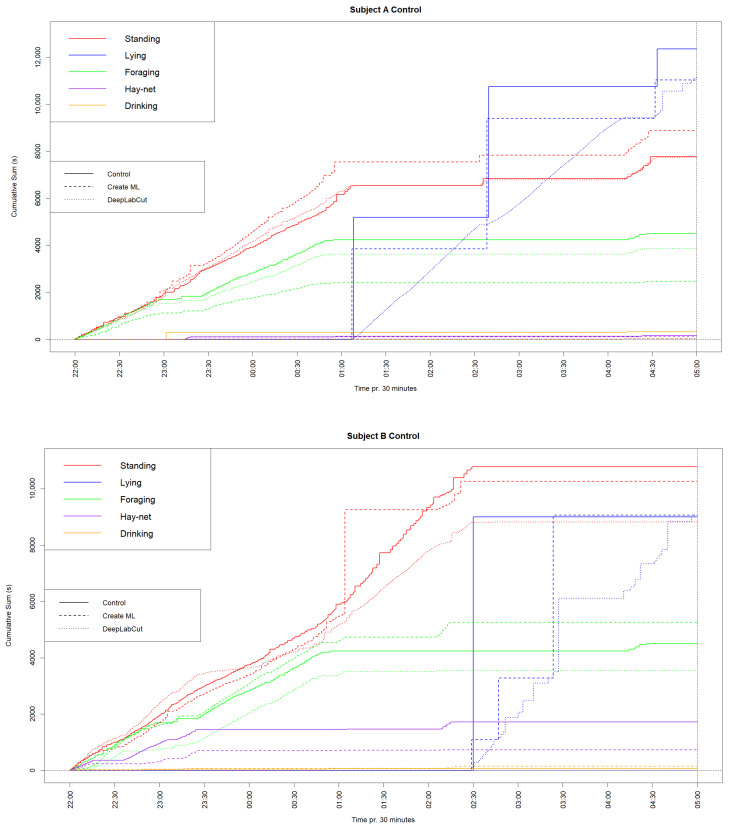
Graph showing the cumulative sums of each behavior for Subjects A and B during the 7 h control period, observed manually, using Create ML and DeepLabCut. Behaviors are distinguished by color and method.

**Figure 3 animals-14-02820-f003:**
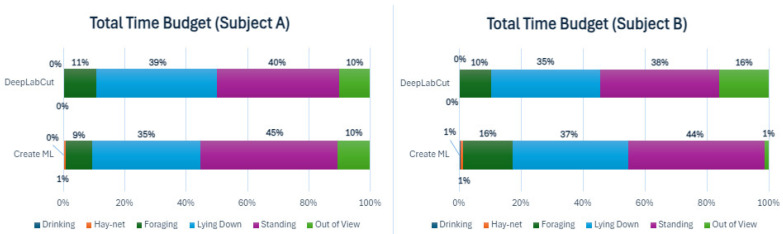
Time budgets for the total time spent on each behavior for Subjects A and B during all seven observed nights, for both ML models. The different colors show different behaviors.

**Figure 4 animals-14-02820-f004:**
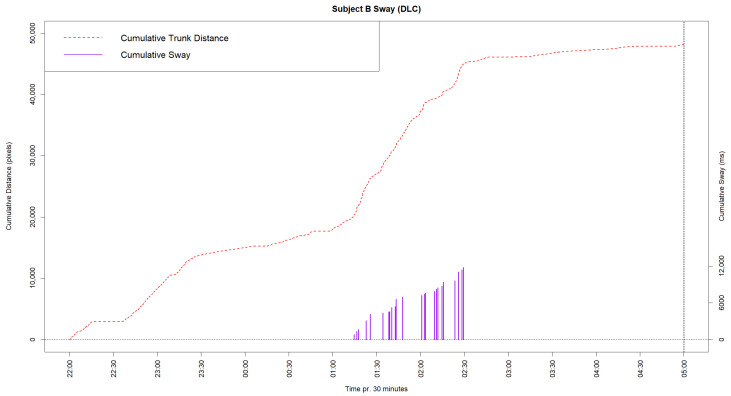
Cumulative graph of trunk root movement (red) and manually observed sway (purple) in the control period. The left *y*-axis shows cumulative pixel movement; the right *y*-axis shows the manually coded sway behavior in milliseconds.

**Table 1 animals-14-02820-t001:** Ethogram used for behavioral coding of the two subjects, used for both manual and automatic coding.

Behavior	Description
Standing	The elephant is standing or walking. This behavior is the default if no other selected behavior is taking place.
Lying down	The elephant is lying down on the floor of the enclosure.
Drinking	The elephant is drinking from a water bowl.
Foraging	The elephant is using the foraging boxes, accessed using trunks in the holes at the back of the enclosure.
Hay-net	The elephant is using its trunk to reach the hay-net at the top of the enclosure.
Swaying	The elephant is swaying from side to side for at least 5 s.
Out of view	The elephant is out of view of the camera. This may also include falsely unlabeled frames by the machine learning models.

## Data Availability

The data presented in this study are available on request from the corresponding author.

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
