# Peer review of "Behavioral Coding of Captive African Elephants (Loxodonta africana): Utilizing DeepLabCut and Create ML for Nocturnal Activity Tracking"

_animals, 2024, doi:10.3390/ani14192820_

Round 1

Reviewer 1 Report

Comments and Suggestions for Authors

Suggested revisions:

Abstract line 20: How was "accuracy" of the models assessed?

Abstract line 22: Should be "... data were...", not "... data was...".

Introduction - last sentence of section 1.1, lines 42-43: This sentence starts in the singular tense, but then switches into the plural tense on line 42 -- "are" should be "is". On line 43, "African elephants" should be in the singular, not plural, tense, and be preceded by "the"... i.e., "... the African elephant...". Also on line 43, there is another plural to singular tense correction to make here -- "have" should be "has".

Section 1.2, line 48: "influences" should be singular: "influence", and "helps" should also be singular: "help".

Section 1.3, lines 78-79: The wording in the phrase, "... for a variety of purposes has been explored in recent years..." sounds backwards to me. I think clearer phrasing would be something like, "... has been explored in recent years in application to a variety of purposes...".

Section 2.1, line 116: The current wording is simply, "In this study...". Was the study conducted in 2024, or back in 2023? Indicate the year of the study (otherwise reference on lines 140-141 to March 12th -18th and April 16th - 21st are made without any clear context). 

Section 2.2, line 137: You need to correct this flagged error and insert text that reads (something like), "; see Table 1, below.".

Page 4, line 140: The term "data" is the plural of "datum" -- therefore, reference to "data" should always be in the plural tense:, i.e., "The data were collected...".

Section 2.3, line 167: Another flagged error that needs to be corrected. Reference to a Figure or Table appears to missing here.

Page 5, line 184: The phrase, "... the models and control..." should read, "... the models and the control...".

Page 5, line 196, re. the phrase, "... manually observed data...": I think the operative phrase here should be, "... manually recorded data...". 

Page 5, line 204: "distance" should be "distances".

Section 3.1.1., lines 212-213: Another Error is flagged following the phrase, "... illustrated below in...". Insert "Figure 1", so the phrase reads, "... illustrated below in Figure 1." 

Section 3.1.1, lines 219-220: Can the "... relatively small differences... " be described as non-significant differences?

Section 3.1.1, lines 236-237: Another Error is flagged following the phrase on line 236 that reads, "... have been constructed and displayed below in...". Insert "Figure 2", so the phrase reads, "... have been constructed and displayed below in Figure 2.".

Page 8, line 251, re. the closing phrase, "... both models, noticeably for the DeepLabCut model.": Rephrase to read, "... models, most noticeably...".

Section 3.2.1, line 290: Another Error is flagged at the end of the sentence on line 290. Insert "... Figure 3." at the end of the sentence.

Section 3.2.2, line 342: "... were plotted;..." should be "... was plotted; ...".

Section 3.2.2, line 343: Another Error is flagged at the end of the sentence on line 343. Insert "...Figure 4, below." to complete the sentence.

Section 4.1, line 356, re. the phrase,"... manually conducted observations." -- I think "... manuallly recorded observations." would be stronger phrasing.

Section 4.2, line 386, re. the phrase, "... It is however inconclusive whether Subject B..." -- A note on phrasing and punctuation... the term "however" should be 'nested' between commas -- e.g., "It is, however, inconclusive whether Subject B...".

Section 4.2, line 389, re. the phrase, "... it does however appear that..." -- See previous comment re. 'using commas to 'nest' the term "however". 

Section 4.2, line 412 -- "... Asian Elephants..." should be, "... Asian elephants...".

Section 4.2, line 416 -- Replace "which" with "this"; i.e., "... a general circadian rhythm; this is in accordance with...".

 Section 4.2, line 418: Replace "Although..." with "However, ...".

Section 4.2, line 423: re. the phrase, "... result is however in accordance with..." -- Punctuation here -- it should read, ""... result is, however, in accordance with...". 

Section 4.3, line 433, re. "Other applications of machine learning models for behavioral coding": I believe that in a subsection with this heading, at least some general comments/remarks of the applicability (*or lack thereof*) of these ML data coding techniques to studies on *other taxa* in zoo settings where those species may also be prone to demonstrating stereotypical behaviours. For example, various species of large carnivores (ursids, felids, canids) are well-known to often demonstrate apparently stereotypical behaviours (such as repeated "pacing" and "head-tossing"). But, other behaviours that may be indicative of stress and/or stereotypy (such as obsessive self-grooming) may also be very difficult to accurately capture on nocturnal video recordings. At the least, subsection 4.3 on "Other applications of machine learning..." should include a preliminary assessment of both the *known* and *potential* limitations of ML applications in facilitating zoo-based data monitoring of species of concern/interest. For example, body size of the species of interest is clearly a key variable in applying these ML techniques. The potential limitations when working with smaller nocturnal species (such as microtine rodents, or nocturnal reptiles) for the approaches/techniques described in the paper to be usefully applied should be explicitly recognized. But, in contrast to the rodent/reptile limitations just noted, the techniques described in this paper could perhaps be usefully applied to other other species in zoo settings (especially larger Artiodactyl and Perrisodactyl species -- e.g., large bovids, cervids, rhinos, tapirs, etc.). Some indication of the potential for such wider application of techniques such as the sort presented in this paper would be a useful broader statement. 

Section 4.3, lines 439-440: The term "however" should be 'nested' between commas -- i.e., "... this behavior is, however, primarily visual...".

Section 4.3, line 444: A friendly editorial suggestion here; I suggest replacing "the animal" with "individuals"; alternatively, "the animal" could also be replaced with "individuals animals". That is: "... gain insight into the welfare of individuals."; or,  "... gain insight into the welfare of individual animals." 

Section 5, line 451: "7" should be "seven".

Section 5, lines 454-455 -- re., "... further expansion and adjustments may be desired.": See my comment above about the need for describing/outlining what you see as the *limits of applicability* for the techniques you have assessed.

p. 16, re. the heading "Figure 4. A.": Format for this heading should be "Figure A4." in order to be internally consistent with the heading format of the other Figures. 

p. 19, 'References' -- re. ref. 10, "(Elephas maximus)" should be italicized: "(Elephas maximus)".

p. 19,  'References' -- re. ref. 16, "(Loxodonta africana)" should be italicized: "(Loxodonta africana)". 

p. 20,  'References' -- re. ref. 18, (Loxodonta africana)" and "(Elephas maximus)" should both be italicized: "(Loxodonta africana)" and "(Elephas maximus)".

p. 20, 'References' -- re. ref. 27, "(Elephas maximus)" should be italicized:                          "(Elephas maximus)".

p. 20,  'References' -- re. ref. 28, "(Loxodonta africana)" should be italicized: "(Loxodonta africana)".

p. 20,  'References' -- re. ref. 33, "(Canis familiaris)" should be italicized: "(Canis familiaris").

p. 21,  'References' -- re. ref. 44, "(Loxodonta africana)" should be italicized: "(Loxodonta africana)".

p. 21, 'References' -- re. ref. 52, "(Elephas maximus)" should be italicized:                          "(Elephas maximus)"; "Zoological and Botanical Gardens" should be: "Journal of Zoological and Botanical Gardens".

p. 21,  'References' -- re. ref. 53, "(Loxodonta africana)" should be italicized: "(Loxodonta africana)".

Comments on the Quality of English Language

See my comments above concerning grammatical and editorial revisions.

Author Response

Thank you for your valuable feedback! We have revised the paper to fulfill these requests.

1)

"Abstract line 20: How was "accuracy" of the models assessed?

Abstract line 22: Should be "... data were...", not "... data was..."."

These issues in the abstract have been fixed.

2)

"Introduction - last sentence of section 1.1, lines 42-43: This sentence starts in the singular tense, but then switches into the plural tense on line 42 -- "are" should be "is". On line 43, "African elephants" should be in the singular, not plural, tense, and be preceded by "the"... i.e., "... the African elephant...". Also on line 43, there is another plural to singular tense correction to make here -- "have" should be "has".

Section 1.2, line 48: "influences" should be singular: "influence", and "helps" should also be singular: "help".

Section 1.3, lines 78-79: The wording in the phrase, "... for a variety of purposes has been explored in recent years..." sounds backwards to me. I think clearer phrasing would be something like, "... has been explored in recent years in application to a variety of purposes..."."

The issues with the tense and other language issues have been fixed.

3)

"Section 2.1, line 116: The current wording is simply, "In this study...". Was the study conducted in 2024, or back in 2023? Indicate the year of the study (otherwise reference on lines 140-141 to March 12th -18th and April 16th - 21st are made without any clear context). 

Section 2.2, line 137: You need to correct this flagged error and insert text that reads (something like), "; see Table 1, below.".

Page 4, line 140: The term "data" is the plural of "datum" -- therefore, reference to "data" should always be in the plural tense:, i.e., "The data were collected...".

Section 2.3, line 167: Another flagged error that needs to be corrected. Reference to a Figure or Table appears to missing here.

Page 5, line 184: The phrase, "... the models and control..." should read, "... the models and the control...".

Page 5, line 196, re. the phrase, "... manually observed data...": I think the operative phrase here should be, "... manually recorded data...". 

Page 5, line 204: "distance" should be "distances"."

The time of the study has been noted along with the dates. All of the flagged errors should also be fixed. The grammatical errors have been fixed.

4)

"Section 3.1.1., lines 212-213: Another Error is flagged following the phrase, "... illustrated below in...". Insert "Figure 1", so the phrase reads, "... illustrated below in Figure 1." 

Section 3.1.1, lines 219-220: Can the "... relatively small differences... " be described as non-significant differences?

Section 3.1.1, lines 236-237: Another Error is flagged following the phrase on line 236 that reads, "... have been constructed and displayed below in...". Insert "Figure 2", so the phrase reads, "... have been constructed and displayed below in Figure 2.".

Page 8, line 251, re. the closing phrase, "... both models, noticeably for the DeepLabCut model.": Rephrase to read, "... models, most noticeably...".

Section 3.2.1, line 290: Another Error is flagged at the end of the sentence on line 290. Insert "... Figure 3." at the end of the sentence.

Section 3.2.2, line 342: "... were plotted;..." should be "... was plotted; ...".

Section 3.2.2, line 343: Another Error is flagged at the end of the sentence on line 343. Insert "...Figure 4, below." to complete the sentence."

The grammatical errors and flagged reference errors have been fixed.

5)

"Section 4.1, line 356, re. the phrase,"... manually conducted observations." -- I think "... manuallly recorded observations." would be stronger phrasing.

Section 4.2, line 386, re. the phrase, "... It is however inconclusive whether Subject B..." -- A note on phrasing and punctuation... the term "however" should be 'nested' between commas -- e.g., "It is, however, inconclusive whether Subject B...".

Section 4.2, line 389, re. the phrase, "... it does however appear that..." -- See previous comment re. 'using commas to 'nest' the term "however". 

Section 4.2, line 412 -- "... Asian Elephants..." should be, "... Asian elephants...".

Section 4.2, line 416 -- Replace "which" with "this"; i.e., "... a general circadian rhythm; this is in accordance with...".

 Section 4.2, line 418: Replace "Although..." with "However, ...".

Section 4.2, line 423: re. the phrase, "... result is however in accordance with..." -- Punctuation here -- it should read, ""... result is, however, in accordance with...". 

Section 4.3, line 433, re. "Other applications of machine learning models for behavioral coding": I believe that in a subsection with this heading, at least some general comments/remarks of the applicability (*or lack thereof*) of these ML data coding techniques to studies on *other taxa* in zoo settings where those species may also be prone to demonstrating stereotypical behaviours. For example, various species of large carnivores (ursids, felids, canids) are well-known to often demonstrate apparently stereotypical behaviours (such as repeated "pacing" and "head-tossing"). But, other behaviours that may be indicative of stress and/or stereotypy (such as obsessive self-grooming) may also be very difficult to accurately capture on nocturnal video recordings. At the least, subsection 4.3 on "Other applications of machine learning..." should include a preliminary assessment of both the *known* and *potential* limitations of ML applications in facilitating zoo-based data monitoring of species of concern/interest. For example, body size of the species of interest is clearly a key variable in applying these ML techniques. The potential limitations when working with smaller nocturnal species (such as microtine rodents, or nocturnal reptiles) for the approaches/techniques described in the paper to be usefully applied should be explicitly recognized. But, in contrast to the rodent/reptile limitations just noted, the techniques described in this paper could perhaps be usefully applied to other other species in zoo settings (especially larger Artiodactyl and Perrisodactyl species -- e.g., large bovids, cervids, rhinos, tapirs, etc.). Some indication of the potential for such wider application of techniques such as the sort presented in this paper would be a useful broader statement. 

Section 4.3, lines 439-440: The term "however" should be 'nested' between commas -- i.e., "... this behavior is, however, primarily visual...".

Section 4.3, line 444: A friendly editorial suggestion here; I suggest replacing "the animal" with "individuals"; alternatively, "the animal" could also be replaced with "individuals animals". That is: "... gain insight into the welfare of individuals."; or,  "... gain insight into the welfare of individual animals." "

All the grammatical errors in the discussion have been fixed. Section 4.3 has been expanded to include a paragraph regarding other potentially challenging behavior and some literature about possible solutions to similar issues. 

6)

"Section 5, line 451: "7" should be "seven".

Section 5, lines 454-455 -- re., "... further expansion and adjustments may be desired.": See my comment above about the need for describing/outlining what you see as the *limits of applicability* for the techniques you have assessed."

The conclusion has been revised and expanded to address potential challenges and possible solutions and applications of future studies. 

7)

"p. 16, re. the heading "Figure 4. A.": Format for this heading should be "Figure A4." in order to be internally consistent with the heading format of the other Figures. 

p. 19, 'References' -- re. ref. 10, "(Elephas maximus)" should be italicized: "(Elephas maximus)".

p. 19,  'References' -- re. ref. 16, "(Loxodonta africana)" should be italicized: "(Loxodonta africana)". 

p. 20,  'References' -- re. ref. 18, (Loxodonta africana)" and "(Elephas maximus)" should both be italicized: "(Loxodonta africana)" and "(Elephas maximus)".

p. 20, 'References' -- re. ref. 27, "(Elephas maximus)" should be italicized:                          "(Elephas maximus)".

p. 20,  'References' -- re. ref. 28, "(Loxodonta africana)" should be italicized: "(Loxodonta africana)".

p. 20,  'References' -- re. ref. 33, "(Canis familiaris)" should be italicized: "(Canis familiaris").

p. 21,  'References' -- re. ref. 44, "(Loxodonta africana)" should be italicized: "(Loxodonta africana)".

p. 21, 'References' -- re. ref. 52, "(Elephas maximus)" should be italicized:                          "(Elephas maximus)"; "Zoological and Botanical Gardens" should be: "Journal of Zoological and Botanical Gardens".

p. 21,  'References' -- re. ref. 53, "(Loxodonta africana)" should be italicized: "(Loxodonta africana)"."

The appendix and references have been fixed.

We hope this revision will be adequate, and we look forward to further cooperation.

Best regards.

Reviewer 2 Report

Comments and Suggestions for Authors

Overall:  This paper is very well-written and provides evidence to support the use of AI and machine learning to help with behavioral analysis of captive animals. 

Abstract and Simple Summary:  The abstract and simple summary are both clear with regards to providing sufficient background and justification as well as details about the study methods, findings, and applications. 

Introduction:  Overall, the introduction is good and provides good background literature for both the machine learning component as well as the elephant behavioral component.   The introduction is set up with the first subsection focusing on elephant behavior and the second on machine learning.  The aims are set up so the machine learning component appears first.  I would suggest that the aims should be flipped to mirror the order of the subsections in the introduction.

Materials and Methods:  Overall the Materials and Methods section is good and provides a thorough explanation of the methods.  There should be additional details regarding the training of the human coding methods and the background/credentials of those individuals chosen to manually code. 

Line 137 has “Error! References source not found.” Which needs to be replaced with a reference. 

Line 167 – “Error! References source not found.” Needs to be replaced.

Line 213 - “Error! References source not found.” Needs to be replaced.

Lines 236-235 - “Error! References source not found.” Needs to be replaced.

Line 290 - “Error! References source not found.” Needs to be replaced.

Line 343 - “Error! References source not found.” Needs to be replaced.

Discussion and Conclusion:  Both the discussion and conclusion were thorough and addressed all the major points of the results section and contextualized them in existing literature.  Since there isn’t much research in using AI and machine learning for behavioral analysis of animals (especially those not in lab environments), it’s understandable that there isn’t much existing literature in this section.   

Author Response

Thank you for your kind review! We have revised the comments.

1)

"Introduction:  Overall, the introduction is good and provides good background literature for both the machine learning component as well as the elephant behavioral component.   The introduction is set up with the first subsection focusing on elephant behavior and the second on machine learning.  The aims are set up so the machine learning component appears first.  I would suggest that the aims should be flipped to mirror the order of the subsections in the introduction."

Thank you for this insight. We understand that the aims should line up with the order of the rest of the article. However, currently the aims line up with the order of the results instead of the introduction, which we believe may allow for a better understanding. For this reason, we have decided to let the aims remain unchanged. 

2)

"Materials and Methods:  Overall the Materials and Methods section is good and provides a thorough explanation of the methods.  There should be additional details regarding the training of the human coding methods and the background/credentials of those individuals chosen to manually code."

We have included details about the researchers and the manual coding process in the methods.

3)

Line 137 has “Error! References source not found.” Which needs to be replaced with a reference. 

Line 167 – “Error! References source not found.” Needs to be replaced.

Line 213 - “Error! References source not found.” Needs to be replaced.

Lines 236-235 - “Error! References source not found.” Needs to be replaced.

Line 290 - “Error! References source not found.” Needs to be replaced.

Line 343 - “Error! References source not found.” Needs to be replaced.

All of these error references should be fixed now.

Thank you for your valuable review. We hope this revision will be adequate, and look forward to continuing our cooperation.

Best regards.

Reviewer 3 Report

Comments and Suggestions for Authors

Behavioral Coding of Captive African Elephants (Loxodonta africana): Utilizing DeepLabCut and Create ML for nocturnal activity tracking

1. Very interesting research entitled “Behavioral Coding of Captive African Elephants (Loxodonta africana): Utilizing DeepLabCut and Create ML for nocturnal activity tracking”.

2. The structure of the article complies with the format of the MDPI-animals journal.

3. Table 1 is not referenced. There is an error that says “Error! Reference source not found..” on line 137. The title of table 1 should be at the top.

4. All figures are not referenced. There is an error that says “Error! Reference source not found..” on lines: 137, 167, 213, 237, 290 and 343. Please correct the references.

5. The title of figures 1, 2 and 4 is too long. The title should be short (maximum two lines), explaining the figure in the previous paragraph.

6. 244 frames of the elephants were taken for the field study (line 150).

a)    Was any preprocessing done to crop the image and to correct: noise, high light, low light, shadows, etc.?

b)    Have you tried converting the image to another color space (XYZ, L*a*b*, L*u*v*, HSV, HLS, YCrCb, YUV, I1I2I3, TSL, etc.) to seek more precision in the results?

7. Since the article is about image processing, I suggest you check out the following articles:

·        García-Mateos, G., Hernández-Hernández, J. L., Escarabajal-Henarejos, D., Jaén-Terrones, S., & Molina-Martínez, J. M. (2015). Study and comparison of color models for automatic image analysis in irrigation management applications. Agricultural water management, 151, 158-166.  https://doi.org/10.1016/j.agwat.2014.08.010  

·        Hernández-Hernández, J. L., García-Mateos, G., González-Esquiva, J. M., Escarabajal-Henarejos, D., Ruiz-Canales, A., & Molina-Martínez, J. M. (2016). Optimal color space selection method for plant/soil segmentation in agriculture. Computers and Electronics in Agriculture, 122, 124-132.  https://doi.org/10.1016/j.compag.2016.01.020

8. It is not clear how the elephant images are processed to obtain: Drinking, Hay-net, Foraging, Lying down, Standing, Out of view (figures 1 and 3). I suggest developing a simple algorithm that shows the operation of the DeepLabCut and Create ML models.

9. The conclusions should be expanded and some future work should be considered, taking your research as a basis.

10. Very good bibliography. I hope you can consult more bibliography.

Note: The points mentioned in the review are recommendations from another perspective that gives the authors the opportunity to improve their article. The observations are formulated to propose a different point of view and it is important that the authors address and implement them in their entirety.

Author Response

Thank you for your valuable insight! We have revised the manuscript according to the comments.

1)

"The structure of the article complies with the format of the MDPI-animals journal."

We noticed in the PDF document that the "conflict of interest" statement was marked in red. We were unsure of why this is, since this section is present in the manuscript. If there is an issue with this segment, please let us now, and we will fix it.

2)

"Table 1 is not referenced. There is an error that says “Error! Reference source not found..” on line 137. The title of table 1 should be at the top."

This issue has been fixed.

3)

"All figures are not referenced. There is an error that says “Error! Reference source not found..” on lines: 137, 167, 213, 237, 290 and 343. Please correct the references."

These issues have been fixed.

4)

"The title of figures 1, 2 and 4 is too long. The title should be short (maximum two lines), explaining the figure in the previous paragraph."

We have shortened the descriptions of all these figures, and have attempted to reduce it to two lines, although we found this to be difficult in some cases. We hope the reduced descriptions will still be adequate.

5)

"244 frames of the elephants were taken for the field study (line 150).

a)Was any preprocessing done to crop the image and to correct: noise, high light, low light, shadows, etc.?

b)Have you tried converting the image to another color space (XYZ, L*a*b*, L*u*v*, HSV, HLS, YCrCb, YUV, I1I2I3, TSL, etc.) to seek more precision in the results?"

No preprocessing was done to the images, and this has now been disclosed in section 2.3. We have also not tried converting the images to another color space, which has now been mentioned as a possibility in section 4.1.

6)

"Since the article is about image processing, I suggest you check out the following articles:

  • García-Mateos, G., Hernández-Hernández, J. L., Escarabajal-Henarejos, D., Jaén-Terrones, S., & Molina-Martínez, J. M. (2015). Study and comparison of color models for automatic image analysis in irrigation management applications. Agricultural water management, 151, 158-166.  https://doi.org/10.1016/j.agwat.2014.08.010  
  • Hernández-Hernández, J. L., García-Mateos, G., González-Esquiva, J. M., Escarabajal-Henarejos, D., Ruiz-Canales, A., & Molina-Martínez, J. M. (2016). Optimal color space selection method for plant/soil segmentation in agriculture. Computers and Electronics in Agriculture, 122, 124-132.  https://doi.org/10.1016/j.compag.2016.01.020"

We have read the suggested articles and cited them in section 4.1, where we briefly mention these kinds of possibilities. 

7)

"It is not clear how the elephant images are processed to obtain: Drinking, Hay-net, Foraging, Lying down, Standing, Out of view (figures 1 and 3). I suggest developing a simple algorithm that shows the operation of the DeepLabCut and Create ML models."

We have expanded on this in section 2.3, where we give a more detailed description of the process.

8)

"The conclusions should be expanded and some future work should be considered, taking your research as a basis."

We have expanded the conclusions to include future expansion and possbilities.

9)

"Very good bibliography. I hope you can consult more bibliography."

We have added a total of three new references to the bibliography, including the ones suggested in this review.

Thank you very much for your insightful review! We look forward to continue good cooperation.

Best regards.

Round 2

Reviewer 3 Report

Comments and Suggestions for Authors

Make the following corrections:

1. From point 8, I suggest developing a simple algorithm that shows the operation of the DeepLabCut and Create ML models. 

2. The title of table 1, must be at the top.

3. The format for table 1 is suggested to be the following: (see attached file).

Author Response

Thank you again for your valuable input, and for elaborating on your previously stated comment! We have followed your suggestions as follows:

"1. From point 8, I suggest developing a simple algorithm that shows the operation of the
DeepLabCut and Create ML models. I suggest that the algorithms in this article use the
following format:"

We have followed your suggestion and have created two algorithms, showing the workflow for both DeepLabCut and Create ML. We hope this will be an adequate description of the process.

"2. The title of table 1, must be at the top."

The description of Table 1 has been moved to the top.

"3. The format for table 1 is suggested to be the following: (see attached file)."

We have formatted the table accordingly.

We greatly appreciate your continued cooperation and insightful comments. We hope this revision will be satisfactory.

Best regards, The Authors.